# Inversion and Validation of FY-4A Official Land Surface Temperature Product

Lixin Dong [1,*], Shihao Tang [1], Fuzhou Wang [2], Michael Cosh [3], Xianxiang Li [4,5] and Min Min [4,5]

1   Key Laboratory of Radiometric Calibration and Validation for Environmental Satellites, National Satellite Meteorological Center (National Center for Space Weather), China Meteorological Administration, Beijing 100081, China
2   Hebi Meteorological Bureau of Henan Province, Hebi 458000, China
3   Hydrology and Remote Sensing Laboratory, Agricultural Research Service, USDA, Beltsville, MD 20705, USA
4   School of Atmospheric Sciences, Southern Marine Science and Engineering Guangdong Laboratory (Zhuhai), Sun Yat-sen University (Zhuhai), Zhuhai 519082, China; lixx98@mail.sysu.edu.cn (X.L.); minm5@mail.sysu.edu.cn (M.M.)
5   Guangdong Province Key Laboratory for Climate Change and Natural Disaster Studies, Sun Yat-sen University (Zhuhai), Zhuhai 519082, China
*   Correspondence: donglx@cma.gov.cn

**Abstract:** The thermal infrared data of Fengyun 4A (FY-4A) geostationary meteorological satellite can be used to retrieve hourly land surface temperature (LST). In this paper, seven candidate algorithms are compared and evaluated. The Ulivieri (1985) algorithm is determined to be optimal for the algorithm of FY-4A LST official products. The refined algorithm coefficients for distinguishing dry and moist atmosphere were established for daytime and nighttime, respectively. Then, FY-4A LST official products under clear-sky conditions are produced. The validation results show that: (1) Compared with in-situ measured LST data at the HeBi crop measurement network, the root mean square errors (RMSE) were 2.139 and 2.447 K. Compared with in-situ measured LST data at Naqu alpine meadow site of Tibet plateau, the RMSE was 2.86 K. (2) When compared with the MODIS LST product, the RMSE was 1.64, 2.17, 2.6, and 1.73 K in March, July, October, and December, respectively. By the bias long-time change at a single site, RMSE of the XLHT (city) and GZH (desert) sites were 2.735 and 2.97 K, respectively. Overall, the preferred algorithm exhibits good accuracy and meets the required accuracy of the FY-4A mission.

**Keywords:** FY-4A; geostationary meteorological satellite; AGRI; land surface temperature

## 1. Introduction

LST retrieved by Geostationary Meteorological Satellites (GMS) can be assimilated into climate, weather prediction, and land models for estimating sensible heat and latent heat flux [1,2]. It is also a key factor in the land surface energy budget, which is widely used in meteorology, hydrology, and climate change studies [3]. In recent years, with the capability and efficiency improvement of the new variety of instruments board on GMS, remote sensing with GMS has become an important method for collecting hourly LST at a regional scale [4].

The Fengyun 4A (FY-4A) satellite, launched by China in 2016, has output stable data with remarkable quality [5,6] and shows improved capability and efficiency with high-precision and advanced image registration technology [7]. In order to promote FY-4A operational application, it is necessary to obtain accurate level two (L2) products, such as LST, etc. [8]. The thermal infrared (TIR) channels of Advanced Geosynchronous Radiation Imager (AGRI) design for the FY-4A satellite are similar to MODIS (Moderate Resolution Imaging Spectroradiometer) [9,10]. However, their spectral response functions and spatial resolution are different, the existing land surface temperature (LST) retrieval algorithms

must be evaluated before operational application of the FY-4A TIR. The existing LST algorithms are aimed at different sensors. There are more than twenty existing methods, such as Becker and Li (1990) [11] proposed for Advanced Very High-Resolution Radiometer (AVHRR) data local split window (SW) algorithm. On this basis, Wan and Dozier (1996) [12] further proposed a general SW algorithm for MODIS data. Gillespie et al. (1998) [13] proposed a temperature and emissivity separation algorithm for Advanced Spaceborne Thermal Emission and Reflection (ASTER) multispectral thermal infrared data. Recently, the newly developed TES algorithm for H8/AHI [14] has been used to generate the operational LST product. For these algorithms, different LST inversion models are established by linearization or non-linearity [15].

The development of an operational LST product is one of the main goals of the FY-4A satellite mission, and the accuracy of the LST product (RMSE) is required to be $\pm 2.5$ K. The technical indicators of the LST official product are defined in the FY-4A satellite engineering mission requirements document. In this paper, like GOES-R LST product, seven common SW algorithms, including MODIS product algorithms, are randomly selected and compared to evaluate their applicability for FY-4A TIR. Some of these algorithms are linear or nonlinear, some are sensitive to emissivity, and some are insensitive. The refined algorithm for distinguishing dry and moist atmosphere in daytime and nighttime is developed by radiative transfer simulation and least square fitting. Then, the LST official products are produced and validated.

## 2. Data and Preprocessing

### 2.1. AGRI Data

The AGRI data were obtained from FY-4A project team. The users can download the TIR data with SW channels from the website (http://data.nsmc.org.cn (accessed on 12 April 2023)). In this study, the radiometric correction, geographic correction, and cloud mask were applied to develop the official products. We can get the parameters required for the radiometric corrections from the Scientific dataset of the L1 files. The channel characteristics of FY-4A TIR data are listed in Table 1.

**Table 1.** Channel characteristics of AGRI TIR data.

| Wave-Length (μm) | Sensitivity (300 K) | Spatial Resolution (m) | Temporal Resolution | Main Application |
|---|---|---|---|---|
| 8.0–9.0 | NEΔT ≤ 0.2 K | 4000 | 15 min, Hourly | Cloud, Water vapor |
| 10.3–11.3 (B4) | NEΔT ≤ 0.2 K | 4000 | 15 min, Hourly | Land/water/cloud temperature |
| 11.5–12.5 (B5) | NEΔT ≤ 0.2 K | 4000 | 15 min, Hourly | |

### 2.2. Atmospheric Profiles and WVC Data

There are 7547 cloud-free sounding data in the atmospheric profiles data which from NOAA F98-Weather Products Data Package [4]. Some incorrect atmospheric profile data need to be eliminated. Hence, 143 atmospheric temperature/water vapor profiles were determined according to whether there is cloud or not and whether the data is accurate in this paper. Figure 1 shows the 72 atmospheric profiles for daytime and 71 atmospheric profiles for nighttime. These profiles cover a wide range of atmospheric conditions in the whole year, the column water vapor change from 0.1 to 6.0 g/cm$^2$, and the air temperature change from 230 to 310 K. The latitude range of profiles is from 60° south to 70° north.

In addition, we use the TPW product developed by the product team of FY-4A geo-stationary satellite as the total water vapor content parameter and use the total water vapor content as the input for LST inversion. The FY-4A TPW product can also be downloaded from http://satellite.nsmc.org.cn/PortalSite/Data/Satellite.aspx (accessed on 12 April 2023).

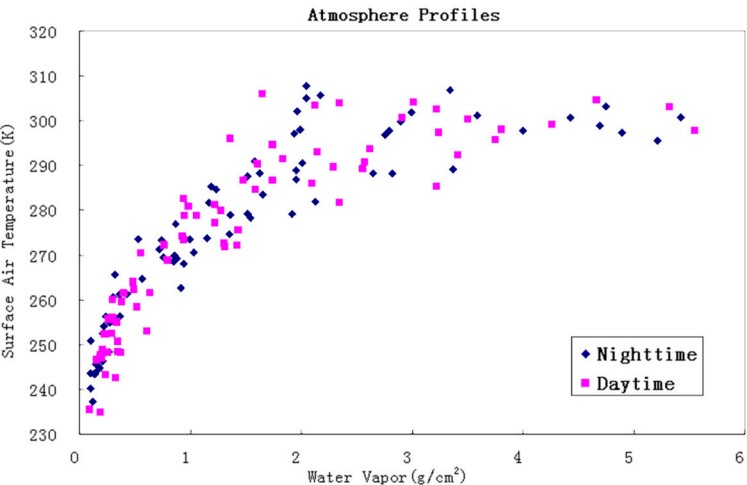

**Figure 1.** Distribution of the selected 143 atmospheric profile data.

### 2.3. Validation Data

There are two independent data for the validation of the FY-4A-LST official products: (1) One year of data from in-situ measurement through two SI-111 infrared radiometers manufactured by Apogee (one aimed at the land surface, another aimed at the sky at a 53° angle), (2) the MODIS LST data provided by NASA, which space-time matching is required for 'Comparative validation'. It can be selected for relative bias analysis due to the accuracy within 1 K by it comparing with ground-measured data [16].

The field-measured in-situ data is recognized as the most ideal authenticity validation data for remote sensing products. As the field measured data, one in-situ site in the Naqu Alpine meadow of the Third Tibetan Plateau Atmospheric Scientific Experiment (TIPEX-III) [17] and six in-situ sites in the Hebi cropland of Henan province are chosen for the validation of FY-4A LST products. These in-situ data are processed by a method described in reference [18] to the true temperature for authenticity validation. The geographic locations of all validation sites are presented in Figure 2 and Table 2. The GZH and XLHT sites are mainly used for comparison with MODIS long-time series products. All sites lie in pure pixels, and the Hebi sites are across two AGRI pixels.

**Table 2.** Validation sites' information.

| Site Name | Code | Latitude | Longitude | Land Cover Type | Data Period |
|---|---|---|---|---|---|
| NaQu | NQ | 92.1212 | 31.817 | Alpine meadow | 2015–2019 |
| HeBi | H26 | 114.482 | 35.669 | Cropland | 2020 May to December |
| | K12 | 114.464 | 35.676 | | 2020 January to December |
| | K14 | 114.487 | 35.674 | | 2020 July to December |
| | K34 | 114.484 | 35.655 | | 2020 April to December |
| | K44 | 114.481 | 35.648 | | 2020 May to December |
| | K54 | 114.481 | 35.638 | | 2020 May to December |
| XiLingHaoTe | XLHT | 116.117 | 43.950 | City | MODIS 2020 |
| GuaiZiHu | GZH | 102.367 | 41.367 | Desert | MODIS 2020–2021 |

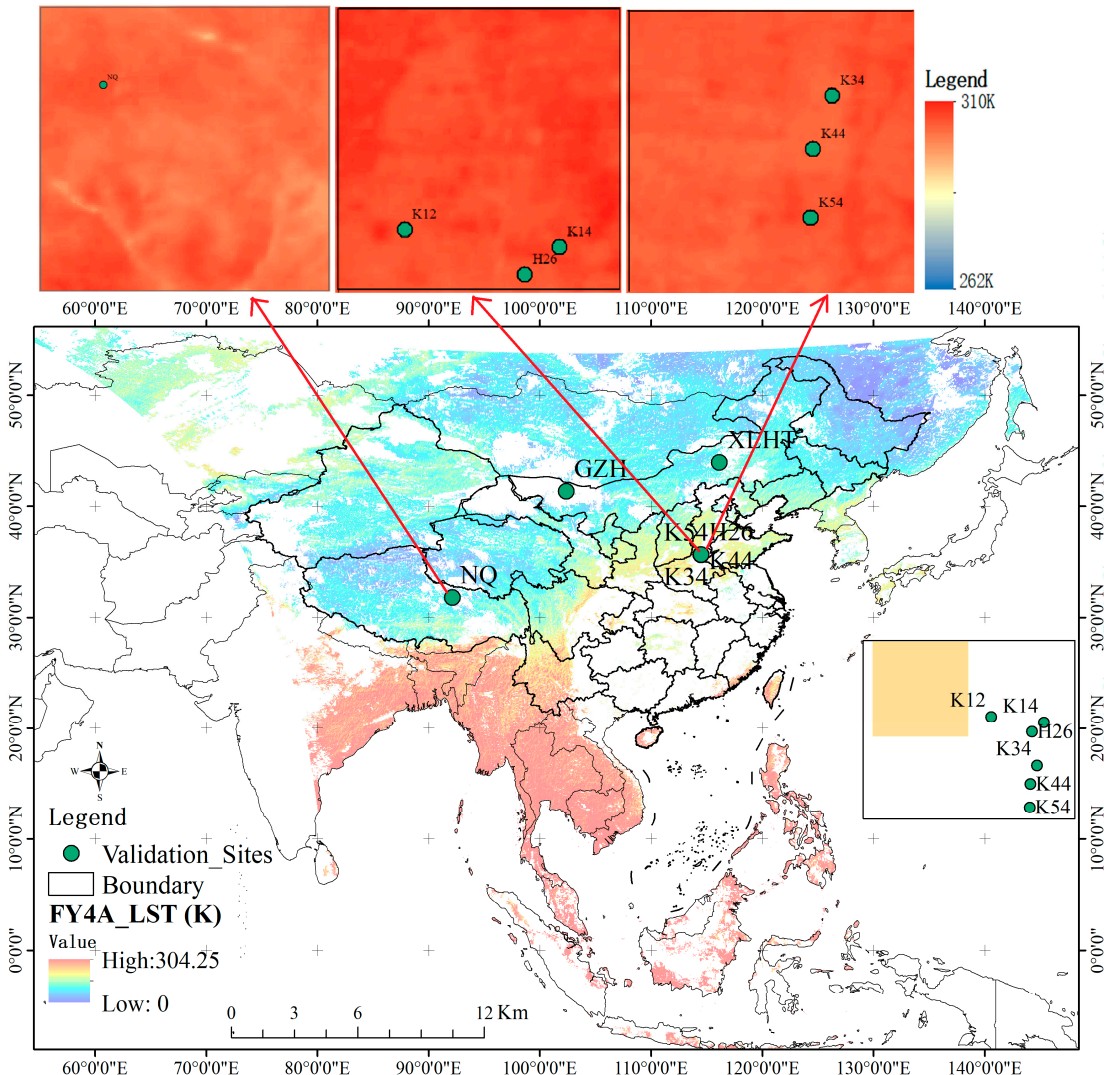

**Figure 2.** Distribution of the validation sites (Down map) and Landsat8 LST (30 × 30 m) data (Up map) matching two AGRI pixels (4 × 4 km) in HeBi (7 June 2020) and NaQu (4 May 2020) in-situ sites; XLHT and GZH are the validation sites with MODIS LST product.

In order to analyze the homogeneity and representativeness of the seven in-situ sites, the 30 m data of Landsat8 LST (https://earthexplorer.usgs.gov/ (accessed on 12 April 2023)) matching with AGRI pixels in these in-situ sites were using respectively to analyze the temporal changes of LST standard deviation (STD) within a year. The up-maps in Figure 2 show the Landsat8 LST on 7 June and 4 May 2020 in HeBi and NaQu sites, respectively. Hebi sites lie mostly distributed in cropland for growing winter wheat and corn. In order to facilitate the construction of sites, six in-situ sites are more than 3–5 m away from rural roads. From May to July, the crops grow luxuriantly, and rural roads are looming.

According to Landsat8 LST statistical result in 2021 (Table 3), the STD within two 4 × 4 km changes from 0.69 to 2.53 K in one year, so Hebi sites are representative and homogeneity in a FY-4A LST pixel. Therefore, it can be used for the validation of FY-4A LST products. However, in Naqu site, the STD within one 4 × 4 km changed from 2.61 to 4.45 K in a year. The maximal STD occurred in April and November, especially. One of the main reasons is the change in land cover caused by snow and stream in autumn and winter, which increases the heterogeneity of pixels. Although STD reached 3.72 k, for wider validation, this site can also be used for the validation of FY-4A LST. On the whole, these sites are representative, and the thermal environment in these pixels is relatively consistent, which can be used for the authenticity validation of FY-4A LST products.

**Table 3.** STD (Unit: K) change within an FY-4A pixel based on Landsat8 LST data.

| Sites/Date | | 2 February 2021 | 19 March 2021 | 20 April 2021 | 22 May 2021 | 7 June 2021 | 26 August 2021 | 11 November 2021 |
|---|---|---|---|---|---|---|---|---|
| **HB** | Pixel1 | 0.7907 | 1.4084 | 1.4555 | 1.5980 | 1.5819 | 0.6915 | 0.9542 |
| | Pixel2 | 1.2384 | 1.8966 | 2.2271 | 2.5338 | 1.9944 | 1.1132 | 0.9447 |
| **Sites/Date** | | 20210213 | 20210301 | 20210415 | 20210504 | / | 20210805 | 20211112 |
| **NQ** | | 3.0933 | 3.3734 | 4.1099 | 2.6176 | / | 3.6166 | 4.4586 |

## 3. Methods

### 3.1. Processing Framework

Figure 3 illustrates the technical flowchart of the main processing steps of the FY-4A LST official products. The candidate algorithms are tested and analyzed using MODTRAN model, and a preferred algorithm is evaluated in step 1. Then, the official algorithm is established in step 2. Finally, the validation and discussion of inversion results based on the preferred FY-4A algorithm are presented in step 3.

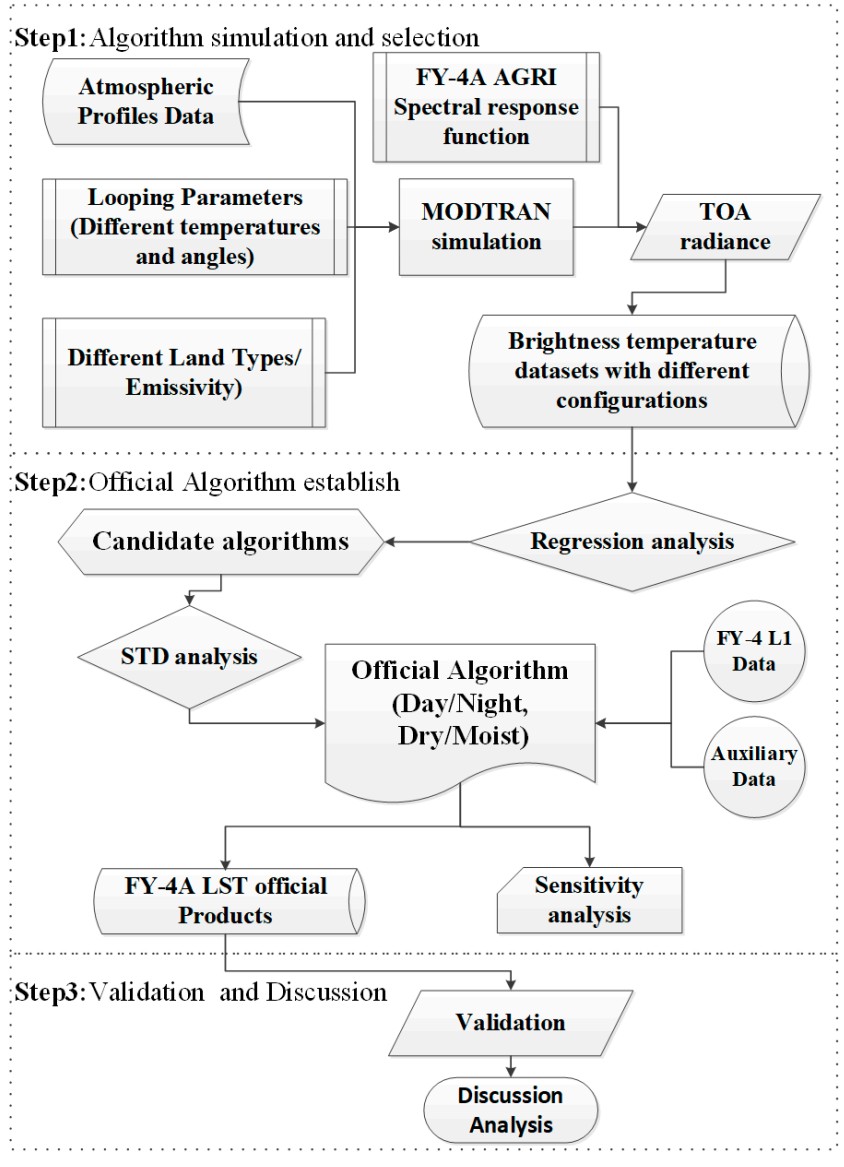

**Figure 3.** The technical flowchart of the FY-4A LST official products. (TOA is Top of Atmosphere).

### 3.2. Algorithms Simulation

The SW algorithm is feasible to retrieve LST [19]. In this study, a preferred SW algorithm for the FY-4A AGRI is determined. The optimal LST algorithm will help to establish the time series datasets for the LST diurnal variation and climate change study [4,20].

#### 3.2.1. Candidate Algorithms

Seven candidate algorithms from the literature (Table 4) are considered and adapted for the AGRI LST inversion [4,12,21–27]. The last term in candidate algorithm is a path length correction [4], which is for further atmospheric correction [28]. This is beneficial to improve the accuracy of the algorithm under a large viewing angle [4]. As we know, McClain et al. (1985) [29] added a viewing angle correction term ($sec\theta - 1$) to the equation. Therefore, the term $(T_4 - T_5)(sec\theta - 1)$ is used for path length correction.

**Table 4.** Candidate SW LST algorithms formulas and reference.

| NO | Candidate Algorithms | Reference of These Algorithm |
|----|----------------------|------------------------------|
| 1 | $T_s = C + A_1 + A_2\frac{1-\varepsilon}{\varepsilon} + A_3\frac{\Delta\varepsilon}{\varepsilon^2}T_4 + T_5 + A_4 + A_5\frac{1-\varepsilon}{\varepsilon} + A_6\frac{\Delta\varepsilon}{\varepsilon^2}T_4 - T_5 + DT_4 - T_5sec\theta - 1$ | *Wan and Dozier 1996* [12] |
| 2 | $T_s = C + A_1\frac{T_4}{\varepsilon} + A_2\frac{T_5}{\varepsilon} + A_3\frac{1-\varepsilon}{\varepsilon} + DT_4 - T_5sec\theta - 1$ | *Prata and Platt 1991* [21] |
| 3 | $T_s = C + A_1T_4 + A_2T_4 - T_5 + A_3T_4 - T_5{}^2 + A_4(1-\varepsilon_4) + A_5\Delta\varepsilon + DT_4 - T_5sec\theta - 1$ | *Coll* et al. *1994* [26] |
| 4 | $T_s = C + A_1T_4 + A_2T_4 - T_5 + A_3\frac{1-\varepsilon}{\varepsilon} + A_4\frac{\Delta\varepsilon}{\varepsilon^2} + DT_4 - T_5sec\theta - 1$ | *Vidal 1991* [23] |
| 5 | $T_s = C + A_1T_4 + A_2T_4 - T_5 + A_3T_4 - T_5\varepsilon_4 + A_4T_5\Delta\varepsilon + DT_4 - T_5sec\theta - 1$ | *Price 1984* [24] |
| 6 | $T_s = C + A_1T_4 + A_2T_4 - T_5 + A_3\varepsilon + DT_4 - T_5sec\theta - 1$ | *Ulivieri and Cannizzaro 1985* [25] |
| 7 | $T_s = C + A_1T_4 + A_2T_4 - T_5 + A_3(1-\varepsilon) + A_4\Delta\varepsilon + DT_4 - T_5sec\theta - 1$ | *Ulivieri* et al. *1992* [22] |

Note: $\varepsilon = \frac{(\varepsilon_4+\varepsilon_5)}{2}$ *and* $\Delta\varepsilon = \varepsilon_4 - \varepsilon_5$. where, $T_4$, $T_5$ is the channel brightness temperature, respectively; $\varepsilon_4$, $\varepsilon_5$ are the emissivity of channels 4 and 5, respectively; and $\theta$ represent the view zenith angle.

#### 3.2.2. Simulation Deploys

In order to select a preferred FY-4A LST algorithm, the algorithm's coefficients were simulated and tested using the MODTRAN model. During simulation, the prescribed LST of each profile changed from 220 to 330 K with a 1-K increment. The sensor view zenith angle was set from 0° to 60°. The pairs were determined using the emissivity data [30]. Upon simulating the top-of-atmosphere radiances, the mean channel radiance was determined by integrating over the FY-4A spectral response function (SRF). The channel radiances were converted into corresponding brightness temperatures using the Planck function. Then, the suitability of those candidate algorithms was analyzed using a simulation dataset.

### 3.3. Official Algorithm Establish

The aforementioned brightness temperature datasets with different configurations were used to establish the algorithm, including regression analysis, estimation of land surface emissivity (LSE), and sensitivity analyses.

#### 3.3.1. WVC Categories

Because the water vapor content (WVC) is one of the most significant parameters in the thermal bands, the brightness temperature data are stratified into two categories according to the total atmospheric WVC. One is the "dry" atmosphere, where the WVC is less than 2.0 g/cm², another is the "moist" atmosphere, where the WVC is larger than 2.0 g/cm². The MODIS LST official algorithm [12] and GOESR LST algorithm [4] were also

using the same stratification. After conducting the least squares analysis with the simulated data, four types of algorithms coefficients were obtained: Dry air conditions during the daytime, dry air condition at night, moist air condition during the daytime, and moist air conditions at nighttime.

### 3.3.2. Calculation of LSE

Here, LSE is calculated by using the *NDVI*-Based Threshold Method. The method is simple and effective and has been applied in many studies [2,30,31].

$$\varepsilon_{i,pixel} = \varepsilon_{i,v}FVC + \varepsilon_{i,g}(1 - FVC) + d\varepsilon_i \tag{1}$$

where, $\varepsilon_{i,v}$ represents the surface emissivity of channel $i$ in the pure vegetation coverage pixel. $\varepsilon_{i,g}$ is the emissivity of channel $i$ in the pure bare ground pixel. When NDVI < 0.2, the pixel is considered to be completely covered by bare soil. At this time, the surface emissivity of the pixel is taken as the reflection value of the soil in the red-light region. The $d\varepsilon_i$ represents the emissivity term produced by multiple reflections of vegetation and the surface for a certain channel. For simplified calculation, assuming that the surface is flat and there is no multiple reflection term, that is $d\varepsilon_i = 0$. *FVC* stands for vegetation coverage and can be calculated from Formula (2):

$$FVC = \frac{NDVI - NDVI_S}{NDVI_V - NDVI_S} \tag{2}$$

where, $NDVI_S$ represents the typical *NDVI* value of pure bare-soil pixels, taking a fixed value of 0.05, $NDVI_V$ represents the typical *NDVI* value of a certain vegetation type in the pure vegetation coverage pixels, and the vegetation cover type is classified by IGBP surface. For each IGBP surface type, $\varepsilon_{i,v}$, $\varepsilon_{i,g}$, $NDVI_V$, and $NDVI_S$ were obtained from published literature data [31–33].

### 3.3.3. Sensitivity Analyses

Two important error sources in LST retrieval are surface emissivity uncertainty and atmospheric water vapor absorption [12].

- LSE Uncertainty

The maximum LST errors $\delta(T_s)$ from the LSE uncertainty can be analytically calculated as:

$$\delta(T_s) = \sqrt{\delta T_1^2 + \delta T_2^2} \tag{3}$$

where $\delta T_1$ and $\delta T_2$ are the 10.8 and 12.0 μm channel errors resulting from the errors of the mean emissivity ($\varepsilon$) and emissivity difference ($\Delta\varepsilon$), respectively. Using algorithm 4 as an example, the LST error attributed to the emissivity error $\delta(T_s)$ can be expressed by the following equation:

$$\delta(T_s) = \sqrt{\left(\left(-\frac{A_3}{\varepsilon^2} - \frac{2A_4\Delta\varepsilon}{\varepsilon^3}\right)\delta\varepsilon\right)^2 + \frac{A_4}{\varepsilon^2}\partial(\Delta\varepsilon)^2} \tag{4}$$

Assuming $\delta\varepsilon_4$ is equal to $\delta\varepsilon_5$, the maximum error is $\delta(\Delta\varepsilon) = |\delta\varepsilon_4| + |\delta\varepsilon_5| = 2\delta\varepsilon$. Thus, the $\delta T_s$ due to the emissivity error can be calculated using Equation (4).

- WVC Uncertainty

The LST algorithm of distinguishing dry and moist atmosphere is established under the condition of the accurate estimation of WVC. However, WVC may be inaccurately calculated due to many error sources in practice. The LST uncertainty attributed to the WVC is primarily caused by using the coefficients of the adjacent WVC subrange [18]. For example, the dry atmospheric algorithm coefficient may be used in the moist atmospheric algorithm, thus obtaining incorrect results. Hence, WVC uncertainty may be tested by

using algorithm coefficients by exchange. Therefore, WVC uncertainty can be assessed with the varying algorithm coefficient using the simulation data sets.

### 3.4. Validation and Analysis

The accuracy of the preferred algorithm needs further validation before the algorithm is used in official products. Here, two indispensable methods are used to the validation of the FY-4A LST products. (1) The 'Authenticity validation' with the in-situ measured data to evaluate the accuracy of the LST product, and (2) 'Comparative validation' to compare the FY-4A LST official product with MODIS LST products. They are very strict in terms of the satellite transit time, view angle, and other terms. The first one is ground truth test if the truth value of pixel scale can be obtained.

## 4. Results and Analysis

### 4.1. Selection of Preferred Algorithm

For determining the preferred FY-4A LST algorithm, we compared the results of the candidate algorithms for different atmospheric conditions. The performance of these independent algorithms is appraised by the algorithm accuracy resulting from MODTRAN simulation. For these algorithms, the bias and standard deviation errors (STDE) are calculated and shown in Figure 4. We can find that the Wan, Vidal, Coll, Price and Ulivieri (1992) algorithms show similar results. The Ulivieri (1992) algorithm with emissivity difference term shows improvement when compared with the Ulivieri (1985) algorithm that uses only an average emissivity correction term.

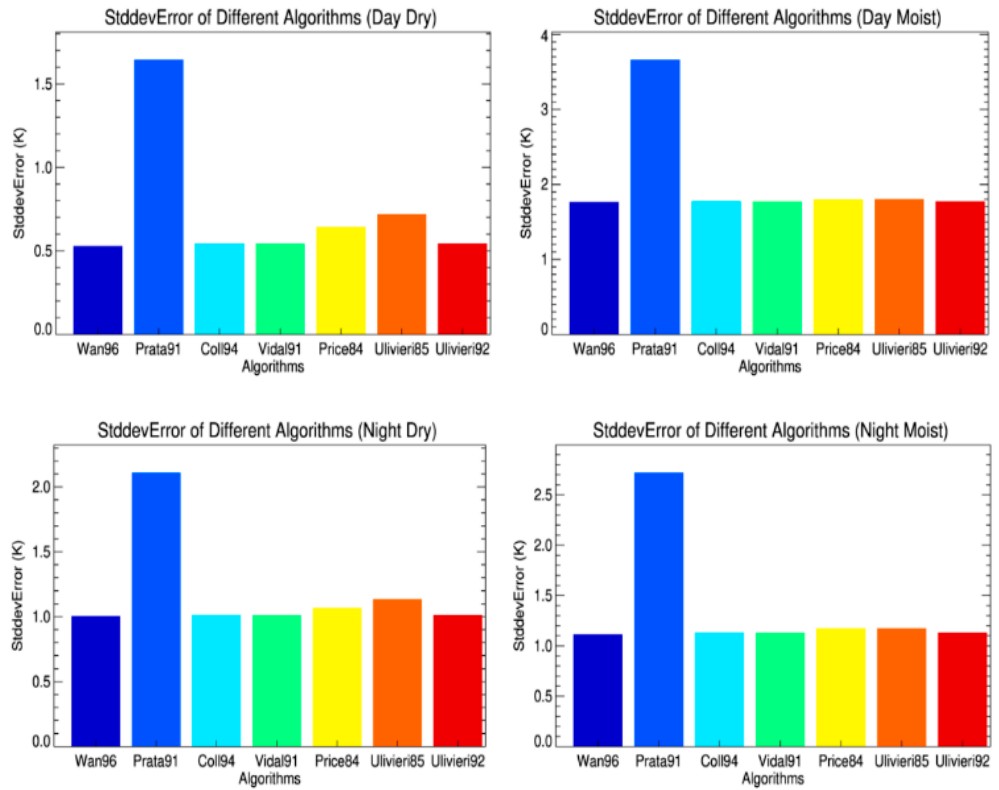

**Figure 4.** STDE of seven candidate algorithms in different conditions deploy.

Figure 5 shows scatter plots for the daytime under dry atmosphere conditions. It indicates that the regression results of all algorithms are good for an LST range from about 250 to 325 K. Excepting the STDE of algorithms Prata algorithm is larger than 1.0, the STDE of the bias between the LST set value and inversion result ranged from 0.529 (algorithms Wan) to 0.720 K (algorithms Ulivieri). For the nighttime algorithms, the STDE changed from 0.795 to 1.137 K (except for the algorithms Prata). Under the moist atmosphere conditions,

the results are similar. Figure 4 shows the STDE of all algorithms. The result of the nighttime under moist atmosphere conditions is similar to the nighttime dry atmosphere cases, however, the STDE of the algorithms for the daytime moist atmosphere cases is not good, which is larger than 1.5 K.

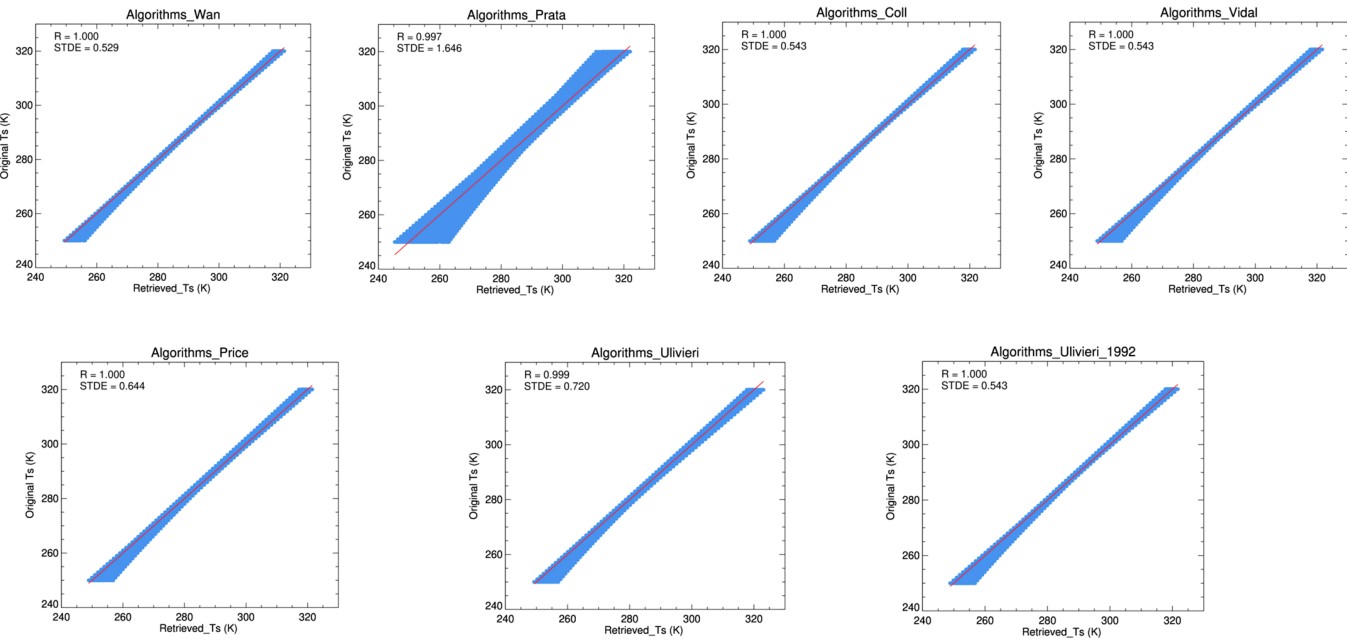

**Figure 5.** Scatter plots of the regression results for the dry atmosphere (daytime).

To analyze bias distributions, the regression bias histogram for the daytime under dry atmosphere conditions are shown in Figure 6. Except for the algorithms *Prata* (bias is between −5 and 5 K), there is no significant bias in many algorithms (bias is between −2 and 2 K). That means not all algorithms performed well. Compared with the result of the daytime algorithm, the STDE at nighttime under the moist atmosphere conditions is slightly worse for each algorithm. This is because the night atmospheric profiles are moister than those of daytime. Under the dry atmosphere conditions, the STDE of each algorithm is quite similar in the daytime and nighttime.

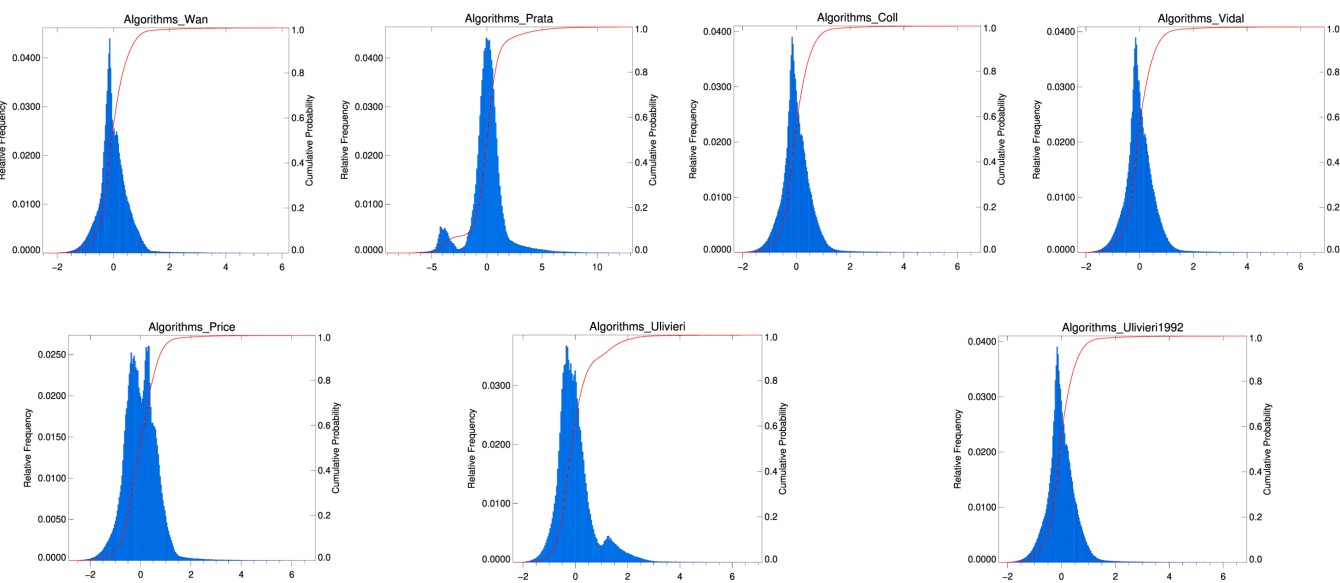

**Figure 6.** The regression bias histogram for the daytime under dry atmosphere conditions.

*4.2. Sensitivity Analysis*

4.2.1. Emissivity Sensitivity Analysis

Figure 7 shows the error changes caused by the emissivity of the surface type, affecting the accuracy of LST inversion algorithms 4 and 6. The algorithm includes four different subclasses: Daytime dry, Daytime moist, Nighttime dry, and Nighttime moist. Assuming that the $\varepsilon$ is equal to 0.97, the $\delta\varepsilon$ is equal to 0.005, and the brightness temperatures of channels 4 and 5 are set to 295 and 294 K, respectively. Of course, the brightness temperature data can also be set to other values, and the change trend is the same.

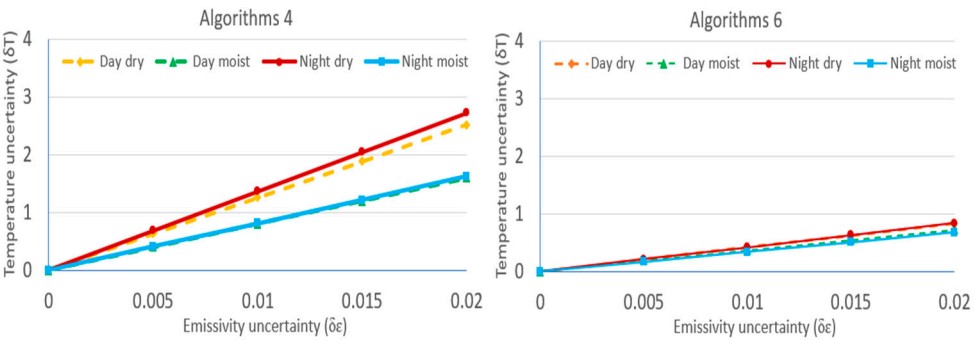

**Figure 7.** Sensitivity of the surface emissivity.

Results show that the LST uncertainty ($\delta T$) increases approximately linearly. Except algorithm 6, the uncertainty can be up to 2−3 K for error 0.02 of emissivity. Moreover, it can be seen that the algorithm of dry atmosphere is higher sensitive to emissivity than that of moist atmosphere whether in daytime and nighttime. Because for dry atmospheres, the LST algorithms are less affected by the atmospheric absorption, its sensitivity to emissivity is the main influencing factor under small WVC conditions.

Moreover, the maximum and minimum range of sensitivity at night is larger than that during the day, it indicates that the sensitivity to emissivity at night is higher for dry atmospheric conditions and lower for moist atmospheric conditions than day cases. Some studies believe that because emissivity exists in mutual cancellation behavior, the final LST error may be significantly smaller practically [4]. The changes of other algorithms are basically the same as those of algorithm 4, which will not be repeated here.

4.2.2. WVC Sensitivity Analysis

Table 5 shows the WVC uncertainty tested by incorrectly applying the LST algorithm coefficient for different atmospheric conditions. It can be seen that for all algorithms, the incorrect selection of the WVC subclass led to a significant RMSE increase in LST retrieval. If these algorithm coefficients derived for dry atmospheric conditions are changed to use for the moist atmospheric algorithm, RMSE increases significantly. For example, if the coefficients of dry air are incorrectly used for the moist atmospheric conditions, the increase of RMSE is greater than that of moist air makes them incorrectly used for the dry atmospheric algorithm.

In a word, WVC sensitivity of all the algorithms is similar. However, algorithms 6 had smaller emissivity sensitivity. Hence, algorithm 6 was chosen for FY-4A LST official product, and its accuracy will be further evaluated and verified in the following section.

As mentioned above, there is no more detailed classification of the water vapor level due to the fear that the error of the WVC itself will lead to a bigger inversion error.

In addition, the sensor noise is also one of the important influence factors. After analyzing the influence of different sensor noises on LST, the inversion error caused by sensor noise is about 0.05–0.15K. The influence of sensor noise on the preferred algorithm is smaller than that of WVC and emissivity. Therefore, the preferred algorithm can obtain a high-precision products to meet the accuracy requirements of the FY-4A mission and further applications.

**Table 5.** RMSE (K) due to incorrectly using coefficients of the different WVC subclass.

| WVC Subclass | | Algorithms | Daytime | | Nighttime | |
|---|---|---|---|---|---|---|
| | | | Dry | Moist | Dry | Moist |
| Daytime | dry | *Uliveri (1985)* | 0.720 | 1.443 | | |
| | | *Wan and Dozier(1996)* | 0.529 | 1.349 | | |
| | | *Vidal (1991)* | 0.543 | 1.418 | / | |
| | moist | *Uliveri (1985)* | 1.923 | 1.8019 | | |
| | | *Wan and Dozier(1996)* | 1.834 | 1.7634 | | |
| | | *Vidal (1991)* | 1.844 | 1.7744 | | |
| Nighttime | dry | *Uliveri (1985)* | | | 1.1365 | 1.558 |
| | | *Wan and Dozier(1996)* | | | 1.0055 | 1.457 |
| | | *Vidal (1991)* | / | | 1.0141 | 1.372 |
| | moist | *Uliveri (1985)* | | | 1.227 | 1.1738 |
| | | *Wan and Dozier(1996)* | | | 1.201 | 1.1157 |
| | | *Vidal (1991)* | | | 1.211 | 1.1301 |

*4.3. Retrieval Results*

The FY-4A brightness temperature data were obtained for the retrieval of FY-4A LST using algorithm 6, which was developed by *Uliveri and Cannizzaro* in 1985. Table 6 gives the algorithm 6 coefficients used for the FY-4A LST official products.

**Table 6.** Preferred algorithm 6 coefficients for FY-4 LST official product.

| Type | C | A1 | A2 | A3 | D |
|---|---|---|---|---|---|
| Daytime dry | 45.258 | 0.985 | 1.332 | −41.750 | 0.035 |
| Daytime moist | 52.651 | 0.931 | 2.408 | −35.962 | −0.219 |
| Nighttime dry | 44.598 | 0.990 | 1.065 | −41.897 | 0.246 |
| Nighttime moist | 61.992 | 0.892 | 2.722 | −33.987 | −0.285 |

Note: Dry/Moist refers to atmospheric conditions when the WVC is less/greater than 2.0 g/cm$^2$.

Figure 8 shows the example of FY-4 LST official product in 2022. On the whole, the temporal variation is reasonable, and the spatial resolution is consistent with our understanding in this area. Based on the brightness temperature data of FY-4A, the cloud and snow detection, and other auxiliary data, FY-4 LST official products with high temporal resolution (every 15 min) under clear-sky condition are produced. Since 2017, the hourly LST products have been released and can be downloaded freely on the website (http://data.nsmc.org.cn (accessed on 12 April 2023)). It will further enhance the refine monitoring capability in many aspects of high-temperature heat wave, urban heat island, and drought disaster.

*4.4. Validation*

The results of simulation validation have been discussed above, in Section 3.1. Next, we discuss the other validation results with the RMSE and the bias spatiotemporal distribution in the following part.

4.4.1. Validation with the In-Situ Data

As Figure 2 and Table 3 show, Hebi and Naqu site data can be used for the diurnal variation of FY-4A LST official products. In Hebi sites, K12, K14, K26 lie in the upper pixel and K34, H44, K54 lie in another lower adjacent pixel. Hence, the average values of K12,

H14, K26 site data and that of K34, H44, K54 site data are respectively used to validate the FY-4A LST products.

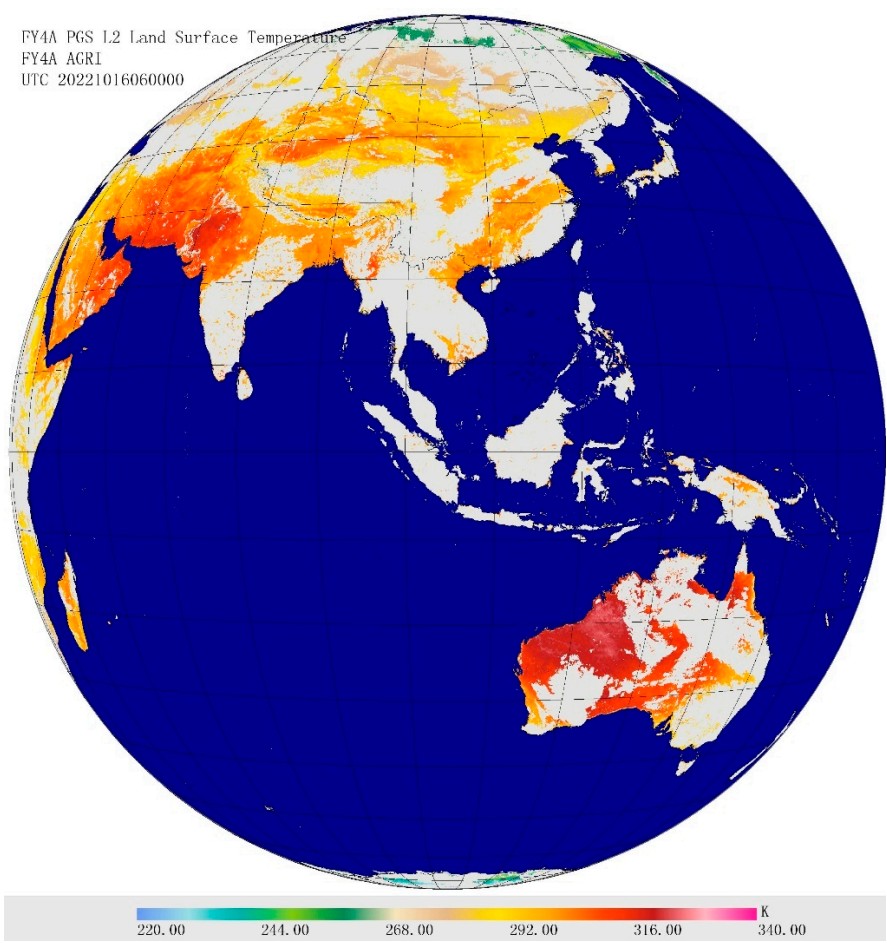

**Figure 8.** FY-4A LST retrieved from AGRI TIR data.

Figure 9 shows the scatter diagrams and error histograms between FY-4A LST and in-situ data based on 2813 and 1776 sample data from 8 January 2020 to 31 December 2020 in Hebi sites, respectively. The RMSE reached 2.447 and 2.139 K in upper pixel and lower adjacent pixel, respectively. According to statistics, in upper pixel, the proportion of data with errors between −2.5 and 2.5 K accounts for 78.315%, and the proportion of data with errors between −3.0 and 3.0 K accounts for 84.963%. In lower pixel, the proportion of data with errors between −2.5 and 2.5 K accounts for 77.871%, and that of sample data with errors between −3.0 and 3.0 K accounts for 85.529%. From the error histogram, it can be found that the values exceeding the error interval are mainly in the negative value area of less than −3.0 K. The possible reason is the influence of cloud pollution or cirrus clouds on LST inversion pixel.

In Naqu site, the 2010 in-situ data from 14 April 2020 to 31 December 2020 are used to validate the FY-4A LST product in Tibet Plateau. Figure 10 shows the scatter diagram and error histogram between FY-4A LST and in-situ data in Naqu site. According to statistics, the RMSE reached 2.86 K and the proportion of sample data with errors between −2.5 and 2.5 K accounts for 71.9%, and the proportion of sample data with errors between −3.0 and 3.0 K accounts for 83.63%. It also can be found from the error histogram that the values exceeding the error interval are mainly in the positive value area larger than 4.0 K. From the error time variation, it can be seen that the positive error area greater than 4.0 K is mainly in the period from 6 October to 7 November 2020. During this period, snow and frost often occur, which makes the ground emissivity incorrectly estimated, resulting in an

error increase. On the other hand, the reason for the negative value area of less than 4.0 K is mainly due to the influence of cloud pollution or cirrus clouds on LST inversion pixel.

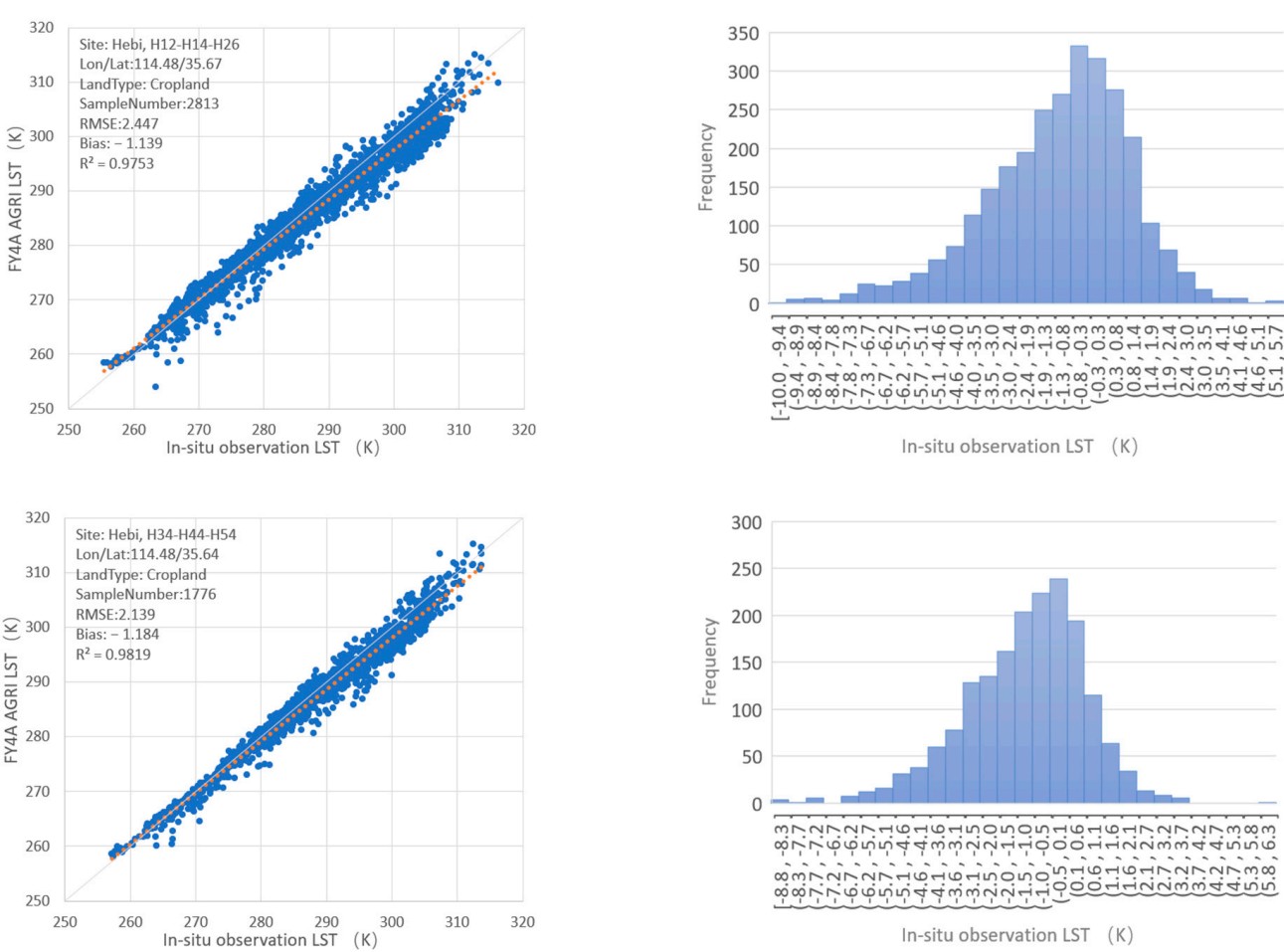

**Figure 9.** Scatter diagram and error histogram between FY-4A LST and in-situ data in Hebi sites.

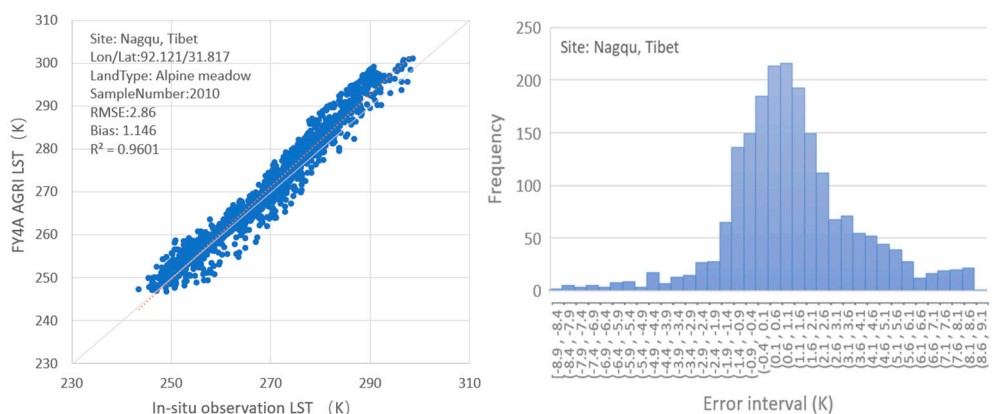

**Figure 10.** Scatter diagram and error histogram between FY-4 LST and in-situ data in Naqu site.

Overall, through the authenticity test based on the in-situ data, the FY-4A LST product accuracy (RMSE) is less than 2.8 K, and the proportion of error within ±3 K can reach about 85%. The remaining 15% of the larger error mainly comes from cloud pollution and the impact of snow and frost weather. These need further research on cloud and snow detection.

### 4.4.2. Comparison with the MODIS LST

Although the bias comparison with MODIS LST products cannot determine the true accuracy of the FY-4A LST official products, it can still reflect the consistency to a certain extent. Therefore, the spatial distribution and long-time changes of the bias between the two products are comprehensively analyzed.

- Bias spatial distribution.

Figure 11 shows the bias spatial distribution between FY-4A LST and MODIS LST products in March, July, October, and December 2018. The 1 km spatial resolution of MODIS LST product is aggregated to the 4 km spatial resolution. The match times between FY-4A LST product and MODIS LST product is 15 min on the main track. It can be seen that the bias of most regions in different seasons is within ± 2.5 K. However, the bias of some regions in July and October exceeded 3 K. This may be the influence of water vapor, which makes the two products inconsistent. Especially in the green triangle boxes of the northwest and northeast corners in October, the overall bias is greater than 3 K, which is mainly due to the greater deviation between the transit time of MODIS and the view time of FY-4A. Statistics show that the RMSE is 1.64, 2.17, 2.6, and 1.73 K in March, July, October, and December, respectively. The correlation coefficient is above 0.9. The mean bias (MB) is −0.53, −1.01, −1.31, and −0.07 K, respectively. For more rigorous validation, you also need to restrict the matching conditions of the view angles. There is no limit to this in this paper. In this sense, the results are satisfactory.

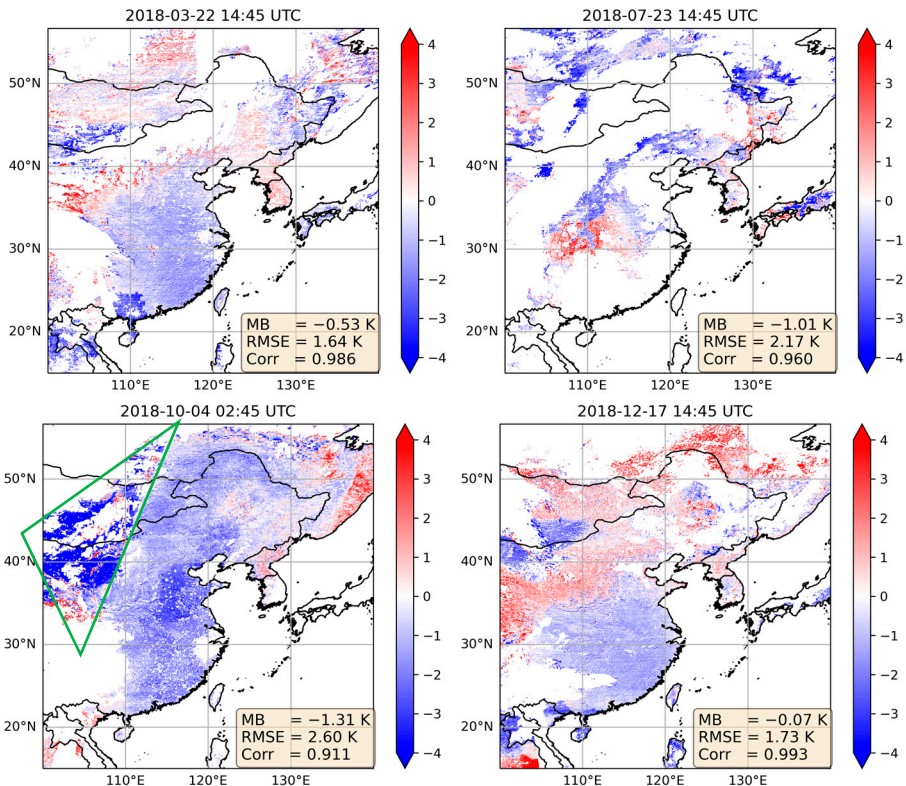

**Figure 11.** Bias spatial distribution map between FY-4 LST and MODIS LST (in the green triangle, the transit time deviation between the MODIS and the FY-4A is larger than 15 min).

- Bias Long-Time Change in Single Site.

The long-time change of bias in single site between FY-4 LST and MODIS LST can reflect the stability of the product. In order to comprehensively study the bias time series changes of two LST official products, we selected two sites with different land cover types in XLHT and GZH to analyze the changes of bias during 8–12 months of 2020 (Figure 12). The 1 km spatial resolution of MODIS LST product is aggregated and matched to the 4 km

spatial resolution, and then the bias between FY-4A LST and MODIS LST is calculated. We can see that the average bias is −0.473 and −2.25 K in XLHT and GZH, respectively. The RMSE is 2.735 and 2.97 K, respectively. Therefore, the stability of the product is good.

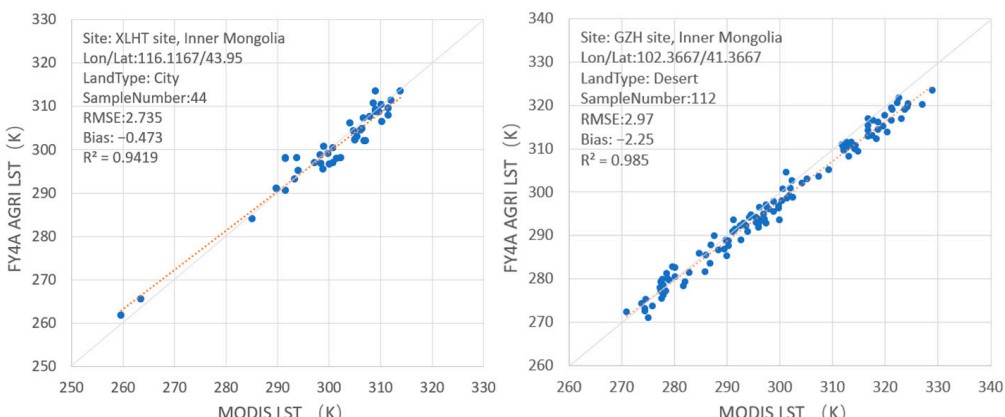

**Figure 12.** Bias long-time change between FY-4 LST and MODIS LST in single site.

## 5. Discussion

### 5.1. Representativeness of the In-Situ Observation

In-situ observations are generally single 'point' measurements, while satellite observations represent a 'surface' measurement of a certain size pixel. FY-4A LST not often represents the result of homogeneous pixel within a 4 × 4 km pixel. Whether the single site observations can represent the average status of the FY-4A pixel is an important problem that needs to be analyzed. Among the site observations used in this paper, the homogeneity of the FY4 pixel at Hebi sites is good, and there are three sites in one pixel, so it has good representativeness. However, because there is only one station, the homogeneity and representativeness of the FY4 pixels in Naqu are relatively poor. More sites need to be further deployed in Naqu to solve the representativeness problems.

The reliability of the in-situ measurements also comes from the positioning accuracy of the FY-4A image and the position matching accuracy of the in-situ sites and the FY-4A image. When matching the in-situ observations to the satellite retrieval result, the FY-4A LST at the closest pixel point is used to match the in-situ measurements. In the satellite and site observation matching process, the spatial difference may also bring some errors.

### 5.2. Cloud Contamination

With the coarse resolution of FY-4A image (4 × 4 km), cloud contamination is an unavoidable problem in LST inversion and validation. To ensure that there are sufficient samples of observation, a 10% cloud threshold is often used. However, due to the different resolutions of the pixels, different results will be formed when determining whether the pixel is in a completely clear-sky state. For example, for FY-4A image, when cloud cover of a pixel is below 10%, it is determined to be a completely clear-sky state (Figure 13). However, MODIS pixels (1 × 1 km) at the same matching position may be cloud pixels. Due to the existence of cloud information, this may have some contamination influence and cause some underestimation of FY-4A LST, while it will occur less often in MODIS clear-sky pixels. Therefore, in the validation process, even if we downscale the MODIS pixel to FY-4A pixel, there will be a problem that FY-4A LST is lower than MODIS LST, unless the FY-4A pixel is under absolutely clear-sky conditions. However, in practice, most of these situations only exist in the northern region in autumn and winter. Hence, cloud contamination is a very important error source in FY-4A LST inversion.

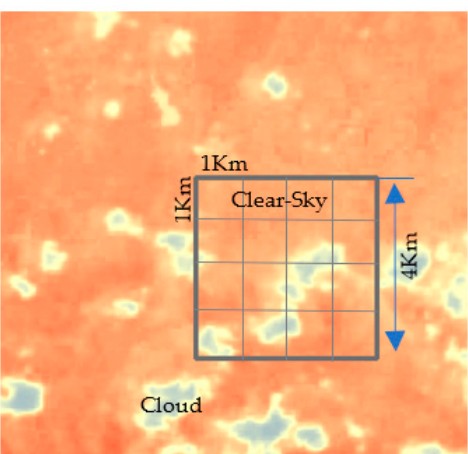

**Figure 13.** Cloud contamination in FY-4A LST pixels (4 × 4 km).

## 6. Conclusions

By using the MODTRAN radiative transfer model, seven candidate LST algorithms are compared to evaluate their applicability for FY-4A TIR data, and then Ulivieri and Cannizzaro (1985) algorithm is determined for the enterprise algorithm of FY-4A LST official products. The refined algorithm in daytime and nighttime under dry and moist atmosphere conditions is developed. Then, the official products of FY-4A LST are produced under clear-sky condition. The validation using one year data of in-situ measurements and MODIS LST product is fulfilled. Results show that the hourly variation of LST can be efficiently obtained, and the accuracy meets the required accuracy of the FY-4A mission.

Of course, it is necessary to further improve the algorithm accuracy. First, emissivity uncertainty is still one of the important influencing factors. Obtaining high-quality LSE is important for a higher accuracy of FY-4A LST official products. Secondly, the validation shows that the FY-4A LST official product accuracy is low in seasons with large atmospheric water vapor. Hence, WVC is another important concern in algorithm development. Although the bright temperature difference can eliminate some water vapor absorption, better water vapor correction methods are needed for LST inversion in high moist atmosphere for FY-4A LST products. Thirdly, how to deal with the impact of cloud contamination is also a very important issue to consider.

Representativeness of the in-situ observation and the reliable verification methodology are necessary for the FY-4A LST products. Cloud contamination filtering, in-situ measurement method fitting with the concept of remote sensing, and scale conversion and matching method of site-to-pixel are all difficult problems in the validation process.

**Author Contributions:** Conceptualization, L.D. and S.T.; methodology, L.D.; validation, L.D., X.L. and M.M.; formal analysis, L.D.; investigation, X.L.; resources, F.W.; data curation, L.D., S.T. and F.W.; writing—original draft preparation, L.D. and M.C.; writing—review and editing, L.D. and M.C. All authors have read and agreed to the published version of the manuscript.

**Funding:** This research was funded by CMA Special Fund for Scientific Research in the Public Interest, The Third Tibetan Plateau Atmospheric Scientific Experiment (TIPEX-III) (GYHY201406001-01).

**Institutional Review Board Statement:** Not applicable.

**Informed Consent Statement:** Not applicable.

**Data Availability Statement:** The FY-4A products have been released and can be downloaded freely on the website (http://data.nsmc.org.cn). The Modis products can be downloaded freely on the NASA website. And the Landsat8 LST can be downloaded freely on https://earthexplorer.usgs.gov/ (accessed on 7 March 2023).

**Acknowledgments:** The authors would like to thank the China National Satellite Center and National Aeronautics and Space Administration for providing the FY-4A and MODIS data, respectively. USDA is an equal opportunity provider and employer.

**Conflicts of Interest:** The authors declare no conflict of interest.

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
