# Peer review of "Inversion and Validation of FY-4A Official Land Surface Temperature Product"

_remotesensing, doi:10.3390/rs15092437_

Round 1

Reviewer 1 Report

Please refer to the attachment for detailed comments.

Author Response

(1) The introduction section lacks a good explanation of why different split-window algorithms are being compared, and only provides a simple list of references. In fact, many researchers have already conducted a significant amount of comparison and analysis. The authors need to supplement the differences between different split-window algorithms and their applicability in the introduction section.

Answer: Modified! Like GOES-R LST product, seven common SW LST algorithms including MODIS product algorithms are randomly selected and compared to evaluate their applicability for FY-4A TIR. Some of these algorithms are linear or nonlinear, some are sensitive to emissivity, and some are insensitive.

(2) Page 4, line 123. “However, the average STD reached 3.72k, and this site can also be used for the validation of FY-4A LST.” Why?

Answer: Modified! Although STD reached 3.72k, for wider validation, this site can also be used for the validation of FY-4A LST.

(3) Page 2, Line 81. Please supplement the reference literature or data download link for FY-4A TPW product, or briefly introduce the inversion algorithm for FY-4A TPW product.

Answer: Modified! The FY-4A TPW product can also be downloaded from http://satellite.nsmc.org.cn/PortalSite/Data/Satellite.aspx.

(4) Page 6, Table 4. Why were the following seven algorithms selected and what was the basis for their selection? Please provide additional information.

Answer: Modified and supplemented in the introduction section.! Like GOES-R LST product, seven common SW LST algorithms including MODIS product algorithms are randomly selected and compared to evaluate their applicability for FY-4A TIR. Some of these algorithms are linear or nonlinear, some are sensitive to emissivity, and some are insensitive.

(5) Page 7, equation (4). What’s the meaning of , the error of LST duo to the uncertainty of LSE for the fourth candidate SW algorithm in Table 4?

Answer:  is Modified to  . The error of LST duo to the uncertainty of LSE for any candidate SW algorithm in Table 4. Not only the fourth candidate SW algorithm. In this article, sensitivity analysis is only conducted on the individual selected algorithm.

(6) The images in the article should be exported as original files, rather than screenshots. For

instance, Figure 3. The font type and size in the images should be consistent throughout the

article.

Answer: Figure 3 and the font type and size in the images were Modified.

(7) Page 6, line 181. The authors should supplement how the bare soil emissivity used in FY-4A LST inversion algorithm is obtained in detail.

Answer: Supplemented! When NDVI<0.2, the pixel is considered to be completely covered by bare soil. At this time, the surface emissivity of the pixel is taken as the reflection value of the soil in the red-light region.

(8) The English in the article needs to be polished.

Answer: Polished by one of the authors, Michael Cosh from USDA.

Reviewer 2 Report

This study developed a practical algorithm for LST retrieval from FY-4A thermal infrared observations. The LST retrieval results were validated by in situ measurements and cross-comparison with a mature LST product-MODIS LST. This study is relevant and with the scope of remote sensing. I recommend publication after minor revision. Also, it is advisable to edit the language by a native English speaker.

Specific comments:

It is suggested to use the title “…FY-4A official land surface temperature product”.

Abstract:

1. line 20, “…in-situ measured LST data at…”

2. line 24, what does “From the bias time change, ” mean?

Introduction:

1. the third paragraph could be more comprehensive, as many LST retrieval algorithms have been proposed. For example, the newly developed TES algorithm for H8/AHI (http://doi.org/10.1109/TGRS.2020.2979846) has been used to generate the operational LST product (https://zenodo.org/record/7578482).

2, line 57, it is advisable to specify the accuracy, bias, rmse or something else.

Data and Preprocessing:

1. line 68, why do atmospheric correction for LST retrieval?

2. Table 1, what does “minutely” mean?

3. Is the meaning of the WVC and TPW the same?

4. Figure 1, the label of x-axis, TPW?

5. Figure 2. Please provide the source of Landsat8 LST.

Methods:

1. what is “enterprise algorithm”.

2. what’s the value of NDVI-v?

Results and analysis:

Section 4.1, I did not find which algorithm was selected and the detailed accuracy of the chosen algorithm.

The quality of Fig. 8 can be improved.

It is better to add bias in the scatter plots (Figs.9 and 10)

What are the time differences between FY-4 and MODIS in Fig. 11.?

Author Response

Specific comments:

 It is suggested to use the title “…FY-4A official land surface temperature product”.

 Answer: Modified

Abstract:

  1. line 20, “…in-situ measured LST data at…”

 Answer: Modified

  1. line 24, what does “From the bias time change,” mean?

  Answer: Modified to “By the bias long-time change at single site,”

Introduction:

  1. the third paragraph could be more comprehensive, as many LST retrieval algorithms have been proposed. For example, the newly developed TES algorithm for H8/AHI (http://doi.org/10.1109/TGRS.2020.2979846) has been used to generate the operational LST product (https://zenodo.org/record/7578482).

 Answer: Modified!

2, line 57, it is advisable to specify the accuracy, bias, rmse or something else.

 Answer: Yes, the (RMSE) is specified.

Data and Preprocessing:

  1. line 68, why do atmospheric correction for LST retrieval?

 Answer: Deleted the atmospheric correction.

  1. Table 1, what does “minutely” mean?

 Answer: Modified to “15 Minute”.

  1. Is the meaning of the WVC and TPW the same?

 Answer: Yes, TPW is Total precipitable water contend.

  1. Figure 1, the label of x-axis, TPW?

 Answer: The label of x-axis is WVC from the atmospheric profiles data.

  1. Figure 2. Please provide the source of Landsat8 LST.

  Answer: Modified and provided. https://earthexplorer.usgs.gov/

Methods:

  1. what is “enterprise algorithm”.

 Answer: The “enterprise algorithm” means “Business Algorithm”.

  1. what’s the value of NDVI-v?

  Answer: NDVIV represents the typical NDVI value of a certain vegetation type in the pure vegetation coverage pixels,

Results and analysis:

Section 4.1, I did not find which algorithm was selected and the detailed accuracy of the chosen algorithm.

 Answer: In Section 4.2.2, algorithm 6 was chosen for FY-4A LST official product.

In a word, WVC sensitivity of all the algorithms is similar. But algorithms 6 had more smaller emissivity sensitivity. Hence, algorithm 6 was chosen for FY-4A LST official product and its accuracy will be further evaluated and verified in following section.

The quality of Fig. 8 can be improved.

 Answer: Modified!

It is better to add bias in the scatter plots (Figs.9 and 10)

 Answer: Modified! The bias was added.

What are the time differences between FY-4 and MODIS in Fig. 11.?

  Answer: Modified! The time differencesbetween FY-4 and MODIS in Fig. 11 is 15 Minute.

Reviewer 3 Report

The manuscript entitled “Inversion and Validation of FY-4A Land Surface Temperature official product” evaluate the sever different algorithms for the evaluation of hourly LST from

 Fengyun 4A (FY-4A) geostationary meteorological satellite. The topic is interesting, it certainly has relevance. The research plan seems good. The methodology is good, but it needs more robust information in the introduction, analysis, and discussion. My major observations are mentioned below:

1.       The research gap and the discussion of the previous literature is needed in the introduction part.

2.       Moreover, the introduction should be organized in such a way to enrich the novelty, which is lacking currently.

3.       A more discussion about the radiometric and all other correction is required. From where you got the parameters required for these corrections.

4.       Figure 3: The left margin arrow mark should be removed.

5.       L141: How did you select a preferred SW algorithm?

6.       Figure 5: Its better to have each figure same dimensions. Same with other figures like 6

7.       Its better to show your numerical values upto two significant digit only.

8.       Do you think validation of spatial predicted LST with the point data is okay? What are the criteria of validation.

9.       What is effect of spatial resolution in validating your 4km LST result with 1km MODIS data.

10.   Discussion is an important part of any study, which is missing in this paper. Current discussion section seems to be limitation of the study. Author needs to include discussion too where they need to discuss their result with the previously published literatures.

Author Response

  1. The research gap and the discussion of the previous literature is needed in the introduction part.

Answer: Modified and supplemented in the introduction section.!

  1. Moreover, the introduction should be organized in such a way to enrich the novelty, which is lacking currently.

Answer: Modified and supplemented in the introduction section.!

  1. A more discussion about the radiometric and all other correction is required. From where you got the parameters required for these corrections.

Answer: Modified! We can get the parameters required for the radiometric corrections from the Scientific dataset of the L1 files

  1. Figure 3: The left margin arrow mark should be removed.

Answer: Modified!

  1. L141: How did you select a preferred SW algorithm?

Answer: In Section 4.2.2, algorithm 6 was chosen for FY-4A LST official product.

In a word, WVC sensitivity of all the algorithms is similar. But algorithms 6 had more smaller emissivity sensitivity. Hence, algorithm 6 was chosen for FY-4A LST official product and its accuracy will be further evaluated and verified in following section.

  1. Figure 5: Its better to have each figure same dimensions. Same with other figures like 6

Answer: Modified!

  1. Its better to show your numerical values upto two significant digit only.

Answer: Modified!

  1. Do you think validation of spatial predicted LST with the point data is okay? What are the criteria of validation.

Answer:  The field measured in-situ data is recognized as the most ideal authenticity validation data for remote sensing products. In order to analyze the homogeneity and representativeness of the seven in-situ sites, the 30 m data of Landsat8 LST matching with AGRI pixels in these in-situ sites were using respectively to analyze the temporal changes of LST standard deviation (STD) within a year.

  1. What is effect of spatial resolution in validating your 4km LST result with 1km MODIS data.

Answer: Maybe is the cloud contamination.

  1. Discussion is an important part of any study, which is missing in this paper. Current discussion section seems to be limitation of the study. Author needs to include discussion too where they need to discuss their result with the previously published literatures.

Answer: Modified!

Reviewer 4 Report

Some important for practical use issues are discussed in the paper presented. The land surface temperature is a key factor for weather predicting and also for evaluating possible heat flux. I have some questions and recommendations to the authors.

1) I do not understand what does the following phrase mean: "... it is necessary to obtain accurate level two products". Does it mean computer programms or something else?

2) Describe, please, the data given in Fig.1. Can You explain, please, the data scatter in this Figure. 

3) I think that Fig. 2 is overloaded by information and is not clear enough.

4) Line 45 should be corrected: "their spectral response functions and resolution is different". May be " are different"?

5) Give the the main advantages and disadvantages of the candidate algorithms, please. Otherwise, one can not imagine "the big picture" of the retrieval process.

6) Line 167 : "Algorithm establish" - should be corrected. 

7) Line 246: "Figure 4 show " - should be corrected. Figure 4 shows (it is  only one Figure).

8) Plots in Fig. 5  need an explanation.

9) Diagramms in Fig.6 are poorly readable.

10) Plots in Fig.12 should be described with some more details. I recommend the authors to reduce the number of pictures in this Figure.

11) Bias spatial distribution maps depicted in Fig. 11 should be described more carefully, step by step.  I do not see overlap between them.  Line 393 should be corrected: "... and then calculate". May be 'calculated"?  

12) I can not infer from the text what refinement of the candidate algorithm is done. In my opinion, this issue should be discussed with more specifics. 

Author Response

1) I do not understand what does the following phrase mean: "... it is necessary to obtain accurate level two products". Does it mean computer programms or something else?

Answer: Modified! "... it is necessary to obtain accurate level two products, such as LST etc. ".

2) Describe, please, the data given in Fig.1. Can You explain, please, the data scatter in this Figure. 

Answer: Modified! Figure 1 shows the 72 atmospheric profiles for daytime and 71 atmospheric profiles for nighttime. These profiles cover a wide range of atmospheric conditions of atmospheric conditions in the whole year, the column water vapor change from 0.1 to 6.0 g/cm2 and the air temperature change from 230 K to 310 K. The latitude range of profiles are from 60◦ south to 70◦ north.

3) I think that Fig. 2 is overloaded by information and is not clear enough.

Answer: Modified!

4) Line 45 should be corrected: "their spectral response functions and resolution is different". May be " are different"?

Answer: Modified!

5) Give the the main advantages and disadvantages of the candidate algorithms, please. Otherwise, one can not imagine "the big picture" of the retrieval process.

Answer: Modified! Like GOES-R LST product, seven common SW LST algorithms including MODIS product algorithms are randomly selected and compared to evaluate their applicability for FY-4A TIR. Some of these algorithms are linear or nonlinear, some are sensitive to emissivity, and some are insensitive.

6) Line 167 : "Algorithm establish" - should be corrected. 

Answer: Modified!

7) Line 246: "Figure 4 show " - should be corrected. Figure 4 shows (it is only one Figure).

Answer: Modified!

8) Plots in Fig. 5 need an explanation.

Answer: Modified! We made an explanation for Fig. 5.” Figure 5 shows scatter plots for the daytime under dry atmosphere conditions. It indicates that……

9) Diagrams in Fig.6 are poorly readable.

Answer: Modified! “To analysis bias distributions, the regression bias histogram for the daytime under dry atmosphere conditions are shown in Figure 6. Except the algorithms Prata (Bias is between 5 and -5), there is no significant bias in many algorithms (Bias is between 2 and -2). “

10) Plots in Fig.12 should be described with some more details. I recommend the authors to reduce the number of pictures in this Figure.

Answer: Modified!

11) Bias spatial distribution maps depicted in Fig. 11 should be described more carefully, step by step.  I do not see overlap between them.  Line 393 should be corrected: "... and then calculate". May be 'calculated"?  

Answer: Modified!

12) I can not infer from the text what refinement of the candidate algorithm is done. In my opinion, this issue should be discussed with more specifics. 

Answer: In Section 4.2.2, algorithm 6 was chosen for FY-4A LST official product.

In a word, WVC sensitivity of all the algorithms is similar. But algorithms 6 had more smaller emissivity sensitivity. Hence, algorithm 6 was chosen for FY-4A LST official product and its accuracy will be further evaluated and verified in following section.

Round 2

Reviewer 3 Report

1. Each sub figure of Figures 5 and 6 should have equal dimensions to look better.

2. Can it possible to show FIgure 8 in UTM or WGS projection in 2 dimensions?

3. Disuccison and Analyse should be only the "Discussion" section.

4. In the Figure 11 caption, it should be better to mention about the green triangle. 

5. In Figure 2, the North arrow and scale are missing.

Author Response

  1. Each sub figure of Figures 5 and 6 should have equal dimensions to look better.

Answer: Modified!

  1. Can it possible to show Figure 8 in UTM or WGS projection in 2 dimensions?

Answer: Yes. as shown in Figure 2, FY-4A LST can be projected to longitude and latitude, UTM or WGS, and other projections. However, what is displayed here is the official nominal product. Please understand and forgive me!

  1. Disuccison and Analyse should be only the "Discussion" section.

Answer: Modified!

  1. In the Figure 11 caption, it should be better to mention about the green triangle. 

Answer: Modified! (In the green triangle, the transit time deviation between the MODIS and the FY-4A is larger than 15 minutes)

  1. In Figure 2, the North arrow and scale are missing.

Answer: Modified
